# Position: The Data Provenance–Parametric Divide in Large Language Models

**Kabilan Elangovan** [1 2]  **Jasmine Ong** [3 4]  **Daniel Ting** [1 2 4]

## Abstract

This position paper argues that as Large Language Models (LLMs) increasingly consume synthetic data, parametric representations can no longer serve as reliable witnesses of factual provenance. Current architectures, which treat fluent outputs as implicitly grounded, create a critical epistemic failure mode: systems emit accurate-looking claims with no recoverable lineage to verifiable sources. We advance the position that *referenceability*—the explicit traceability of claims to accessible evidence—must be enforced as a non-negotiable system invariant. Distinct from Retrieval-Augmented Generation (RAG), which enriches generation with external context, we propose a negative safety constraint: in factual settings, no atomic claim should be emitted unless it is evidence-gated by identifiers that entail it; otherwise, the system must abstain. To operationalize this, we introduce a "separation-of-powers" architecture that decouples parametric generation from factual authorization, along with a diagnostic metric—*Parametric Leakage Ratio* (PLR)—to quantify ungrounded factual emissions. We conclude that enforcing a strict provenance–parametric divide is essential to prevent safety certifications from legitimizing unverifiable outputs in high-stakes domains such as healthcare.

## 1. Introduction

Early large language model success relied on an unstated epistemic assumption: *if it's in the weights, it came from somewhere real*. When models were trained primarily on human-authored corpora (Wikipedia, books, news), parametric storage carried implicit grounding—compressed knowledge largely reflected genuine human consensus at the time of training.

**Synthetic data breaks this assumption.**

We do not argue that parametric representations must be literally fact-free. Rather, we argue that any factual claim surfaced to users in high-stakes settings must be traceable to accessible, verifiable sources. As LLM-generated content increasingly floods the web, training regimes emerge in which Model B learns from Model A's outputs, which may contain hallucinations, outdated information, or forms of "data laundering"—errors that acquire the appearance of consensus through repetition. Shumailov et al. (Shumailov et al., 2024) demonstrate in *Nature* that recursive training on model-generated data induces effectively irreversible defects within the synthetic training loop, including the systematic disappearance of distributional tails. Complementary empirical analyses show that recursive self-training leads to measurable performance degradation: for example, the perplexity distribution of OPT-125M worsens markedly across generations when trained on its own outputs, consistent with progressive loss of rare modes and source diversity under parametric compression rather than simple data noise (Shumailov et al., 2024; Dohmatob et al., 2024).

**Scope and primary driver.** Our primary driver is **epistemic provenance collapse**: in synthetic/weakly sourced regimes, parametric generations no longer carry a defensible evidential lineage. Operational update fragility (fine-tuning/editing) and compliance constraints (e.g., deletion/unlearning) act as reinforcing pressures that make provenance-preserving architectures not just preferable but practically necessary in high-stakes contexts.

### 1.1. The Epistemic Gap

The crisis is epistemic, not merely technical: **parametric models cannot distinguish "learned from evidence" from "inherited from synthesis."** Consider three statements of identical fluency:

1. "The Eiffel Tower was completed in 1889" [grounded in historical records]

2. "Hyperactivated antibiotics treat sepsis" [hallucinated medical term]

[1] Singapore Health Services, Singapore [2] Singapore Eye Research Institute, Singapore [3] Singapore General Hospital, Singapore [4] Duke-NUS Medical School, Singapore. Correspondence to: Kabilan Elangovan <kabilan.elangovan@singhealth.com.sg>.

*Proceedings of the $43^{rd}$ International Conference on Machine Learning*, Seoul, South Korea. PMLR 306, 2026. Copyright 2026 by the author(s).

3. "The 2024 election was won by..." [correct but temporally contingent]

Current parametric models treat these identically—as probability distributions over next tokens. They lack architectural mechanisms to encode evidential status, temporal validity, or source reliability. When a model generates Statement 2 with high confidence, we face not just an accuracy problem but a *verification impossibility*: the output carries no recoverable lineage.

### 1.2. Position Statement

**In the synthetic data era, treating parametric outputs as factual claims without explicit referenceability is untenable scientifically, operationally, and legally.** This position has three components:

1. **Architectural:** Referenceability must be enforced where facts are never *surfaced* without retrieved evidence.

2. **Evaluative:** Progress metrics must measure attribution quality alongside accuracy.

3. **Practical:** High-stakes applications require verifiable architectures as a precondition for deployment.

**What is new.** The novelty of this position is not retrieval itself, but the enforcement of an *evidence-gated invariant* that strongly constrains unsourced factual generation at the system-output level. Unlike standard grounded or retrieval-augmented generation, where citations are optional or post-hoc, we argue for architectures in which factual claims cannot be surfaced unless evidence is present; otherwise, the system should abstain or explicitly surface uncertainty.

**This is not "just RAG":** Standard Retrieval-Augmented Generation treats retrieval as an augmentation—a feature that can be toggled on or off to improve performance. We argue for retrieval as a **constraint** to ensure safety. In our proposed architecture, retrieval is not a fallback mechanism; it is a gating function. Our position is stricter: **in high-stakes factual settings, unrestricted parametric factual fallback becomes a safety liability**.

**Scope of guarantees.** We do not claim that evidence-gated systems guarantee truth or eliminate all failure modes. Retrieval pipelines, verifiers, and evidence corpora may themselves contain errors, omissions, temporal inconsistencies, or adversarial manipulations. Our claim is narrower but operationally important: referenceability transforms factual generation from an opaque parametric process into an inspectable and auditable one, where claims can be externally traced, contested, corrected, and versioned.

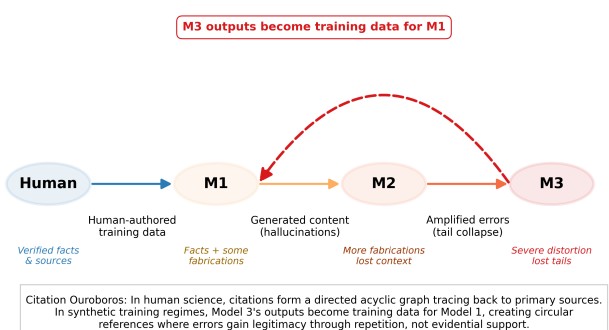

*Figure 1.* **The Citation Ouroboros.** In human science, citations form directed acyclic graphs tracing to primary sources. In synthetic training regimes, outputs become future training data, creating circular references where errors gain legitimacy through repetition rather than evidential support.

## 2. The Synthetic Feedback Loop as Epistemic Failure

### 2.1. Synthetic Amplification and Data Laundering

When model-generated outputs re-enter the training corpus, errors can propagate across generations. Gerstgrasser et al. (Gerstgrasser et al., 2024) show that the critical factor is how synthetic data is integrated: *replacement* of real data with synthetic samples induces model collapse, whereas *accumulation* of synthetic data alongside retained real data preserves stability across generations. However, this stability depends on continued retention of real data, implying ever-growing training corpora and challenging the premise that parametric compression alone can serve as a durable substitute for data provenance.

**Data laundering** occurs when:

- Errors from Model A become "consensus" when Model B trains on A's outputs

- Source identity, timestamp, and authorship are destroyed

- Hallucinations gain probability mass through repetition

Data contamination analyses reveal the scope of benchmark leakage in contemporary language model evaluation. Yang et al. (Yang et al., 2023) report that approximately 8–18% of the HumanEval benchmark overlaps with large-scale pretraining datasets such as RedPajama-Data-1T and StarCoder-Data, including subsets derived from GPT-3.5/4-generated synthetic data. In the absence of explicit provenance tracking, distinguishing genuine learned generalization from synthetic or duplicated training artifacts becomes increasingly unreliable.

## 2.2. Why Correctness Is No Longer Sufficient

A statement can be:

- **Correct but unsupported:** "Aspirin reduces heart attack risk" without citing trials
- **Correct but outdated:** "COVID-19 vaccines are in Phase III trials" (true in 2020, false later)
- **Accidentally correct:** right answer, wrong reasoning

Empirical studies confirm that correctness and groundedness diverge: a non-trivial fraction of factually correct answers remain ungrounded with respect to the provided evidence (Stolfo, 2024). Even retrieval-augmented generation (RAG) systems frequently fail to supply explicit supporting references for all claims in multi-claim responses, particularly in clinical settings (Wu et al., 2025). In high-stakes applications, **verifiability dominates**: the basis for a claim is as important as the claim itself.

Kalai and Vempala (Kalai & Vempala, 2024) proved a statistical lower bound on hallucination rates for calibrated language models, governed by the Good–Turing estimate (the fraction of "singletons" in training data). This is an information-theoretic limit that does not disappear with scaling, motivating systems that retrieve evidence rather than compress it into weights.

# 3. Why Fine-Tuning Is Structurally Incompatible with Referenceability

## 3.1. Weight Updates Destroy Lineage

This process:

- **Distributes** information across layers non-locally (Dai et al., 2022)
- **Entangles** facts with linguistic patterns and reasoning structures
- **Erases** temporal markers, source citations, and confidence bounds

The reversal curse (Berglund et al., 2024) exposes this structural mismatch: models trained on "A is B" fail to reliably answer "B is A." Clinical reasoning requires multidirectional access (symptoms→diagnosis→treatment→contraindications). Parametric encoding optimizes specific prediction paths; retrieval systems support flexible access patterns by construction.

## 3.2. Staleness and Irreversibility

Knowledge editing studies quantify the practical cost of parametric updates. Hase et al. (Hase et al., 2023) show that causal localization estimates from Causal Tracing are poorly

*Table 1.* **Knowledge Update Comparison**

| Method | Update Cost | Audit Trail |
|---|---|---|
| Fine-tuning | Full retraining | None |
| Knowledge editing | Degrades at $n \approx 10$ | None |
| RAG index update | Fast (index update) | Complete |

correlated with successful factual edits in language models, undermining the reliability of using localization signals to guide targeted weight modifications. Additionally, recent work has demonstrated that parameter-modifying editing methods suffer from degradation under sequential edits, with retention of updated knowledge decreasing markedly as the number of edits increases (Hsueh et al., 2024; Li & Chu, 2024). In domains with frequent knowledge changes—such as drug recalls or clinical guideline updates—gradient-based updating is therefore operationally fragile.

## 3.3. The Compliance Intractability of Parametric Unlearning

**Fine-tuning creates legal liabilities retrieval architectures largely avoid.** GDPR Article 17 establishes the "right to erasure" (Villaronga et al., 2018). For parametric models, machine unlearning remains computationally intractable and difficult to verify at scale (Bourtoule et al., 2021; Thudi et al., 2022; Kurmanji et al., 2023). In contrast, deletion in a retrieval-based architecture can be instantaneous and auditable. Organizations face a binary choice: architect for retrieval-grounded compliance, or accept indefinite legal exposure from unerasable parametric data.

# 4. Referenceability as a First-Class System Requirement

## 4.1. Defining Referenceable Responses

A response is **referenceable** if every *atomic factual claim* is either: (1) supported by an explicit, verifiable source, or (2) explicitly marked as uncertain/unsupported.

This definition is intentionally stricter than "has citations." In high-stakes settings, citations function as *audit hooks*: they must enable an external party (clinician, lawyer, reviewer) to reproduce the epistemic basis of the claim.

**Factual-mode contract (system invariant).** In **Factual Mode**, we propose enforcing the following invariant:

> No atomic factual claim may be emitted unless it is attached to evidence identifier(s) whose cited passage(s) entail the claim under a verifier; otherwise the system should abstain or explicitly surface uncertainty.

This is a *mechanistic* requirement: it is enforceable at

runtime (via gating) and measurable offline (via claim–citation validity). It is also the central "delta" from standard RAG, where retrieval can be bypassed and citations can be retrofitted.

Throughout this paper, we use *factual mode* to denote contexts where claims are decision-relevant and externally audited (e.g., clinical, legal, scientific), which we collectively refer to as *high-stakes* settings.

**Three operational properties.** Referenceability requires: **(i) Granularity (claim-level attribution), (ii) Verifiability (citation fidelity),** and **(iii) Honesty (uncertainty/abstention).** For example, a response describing a drug interaction, dosage, contraindication, and monitoring plan contains multiple claims; citing one guideline at the end is not sufficient. The unit of audit is the claim.

### 4.2. Claim Atomization: What Counts as an Atomic Fact?

The hardest operational question is not "can the model cite," but *what must be cited*. We treat an atomic factual claim as a proposition that could be independently checked against a source without requiring additional unstated assumptions. Examples:

- *Temporal facts:* "Drug X was approved in 2023."

- *Causal/clinical claims:* "Drug X reduces mortality in population Y."

- *Constraint/threshold facts:* "Avoid Drug X when eGFR < 30."

- *Epidemiologic facts:* "Incidence in group Y is Z%."

Non-factual statements (definitions, purely linguistic restatements, planning text) need not be cited, but *any* claim that would change a clinical/legal decision should. While linguistic atomization is an active research area, even conservative, coarse-grained propositional filtering provides a safety guarantee superior to unconstrained parametric generation.

**Limitations of atomization.** We acknowledge that claim decomposition is inherently approximate, particularly for long-form reasoning, implicit assumptions, or claims requiring synthesis across multiple documents. Our proposal therefore does not depend on perfect atomization. Instead, it introduces a practical unit of audit that is substantially safer than treating responses as monolithic unverifiable outputs.

**Reproducible recipe (high-level).** In practice, claim atomization and claim–citation mapping can be implemented with a constrained pipeline: (i) segment responses into candidate propositions (rule-based on punctuation + discourse markers), (ii) filter "factual" propositions using a lightweight classifier (or an LLM constrained to binary labels with examples), (iii) require each factual proposition to carry evidence IDs at generation time, and (iv) validate entailment using a verifier (NLI + sampling-based expert review). Crucially, this shifts the burden from generative correctness (hard) to verification (easier). It is computationally simpler to verify if Passage A entails Claim B (NLI) than to generate Fact B from scratch. Appendix B provides a concrete, reproducible procedure (including what to log and how to score).

### 4.3. Why Referenceability Must Be Structural

Post-hoc citation fails because it reverses the epistemic direction. The system generates a conclusion, then searches for something that looks supportive. This creates predictable failure modes:

**Retrofitted attribution (citation shopping).** The model produces a claim from parametric memory, then retrieves supporting-looking sources. Evidence becomes decorative rather than causal.

**Approximate matching (semantic co-occurrence).** The model cites documents that mention the same entities without entailing the claim.

**Temporal mismatch (staleness masked by citations).** The model generates a training-era claim, then cites sources that do not actually support it "as-of" time.

Bohnet et al. formalize *Attributed Question Answering* as a task in which systems must output an explicit $(a, c)$ pair—an answer and a supporting evidence pointer—and evaluate attribution using human judgments aligned with the AIS criteria (Bohnet et al., 2023). While their experiments demonstrate that retrieve-then-read architectures substantially outperform post-hoc and closed-book alternatives on AIS, the underlying structural effect is made explicit in the AIS study of Rashkin et al. (Rashkin et al., 2023). There, systems with architectural access to evidence achieve dramatically higher attribution scores (**87.9%**) than models without such access (**25.2%**). This gap is not an optimization artifact but reflects a causal ordering in the pipeline: evidence must be retrieved *prior* to generation so that claims are produced conditional on verifiable text. Post-hoc attribution instead attempts to retrofit evidence to already-generated content, systematically failing under AIS-style support criteria.

### 4.4. Threat Model and Scope

Our framework primarily targets unsupported parametric factual generation, provenance ambiguity, synthetic-data amplification, and unverifiable factual synthesis. It assumes that retrieved evidence, while imperfect, remains externally

*Table 2.* **What standard RAG still permits (and our contract forbids).**

| Failure mode in practice | Evidence-gated contract effect |
| --- | --- |
| Parametric fallback when retrieval is weak | Must abstain / label uncertainty |
| Mixed cited + uncited factual spans | Every factual claim must carry evidence IDs |
| Post-hoc "citation shopping" | Generation is conditioned on retrieved evidence; citations are causal |
| Entity co-occurrence citations (non-entailing) | Verifier rejects non-entailing citations |
| Temporal mismatch (stale claim + fresh-looking citation) | Evidence carries timestamps; "as-of" checks supported |

inspectable and versionable.

Importantly, the framework does not guarantee that retrieved evidence itself is correct, exhaustive, unbiased, or immune to adversarial manipulation. Retrieval failures, verifier errors, poisoned corpora, and contested evidence remain important open problems. Our claim is therefore not that evidence-gated systems eliminate factual failure, but that they transform such failures into observable and auditable events rather than opaque parametric behavior.

### 4.5. Failure-Case Taxonomy: Standard RAG vs Evidence-Gated Factual Mode

To make the delta testable, Table 2 summarizes what standard RAG pipelines typically permit and what the factual-mode contract forbids by construction.

### 4.6. Separation of Powers Architecture

We propose a three-tier system:

**Tier 1: Frozen Reasoning Core (Parametric).** A model optimized for linguistic competence, task decomposition, and reasoning patterns. It can be updated to improve reasoning, but it is not treated as a factual store for high-stakes factual mode.

**Tier 2: Dynamic Knowledge Layer (Non-Parametric).** Versioned databases, structured knowledge graphs, and document indices with provenance metadata (authors, date, DOI), supporting rapid updates, temporal queries, and auditable deletions.

**Tier 3: Mandatory Attribution Interface.** A constraint layer that (i) links each atomic factual claim to supporting sources, (ii) surfaces uncertainty when evidence is weak or conflicting, and (iii) enforces abstention when evidence is insufficient, with complete audit logs.

This separation is computationally competitive:

**Separation of Powers Architecture**

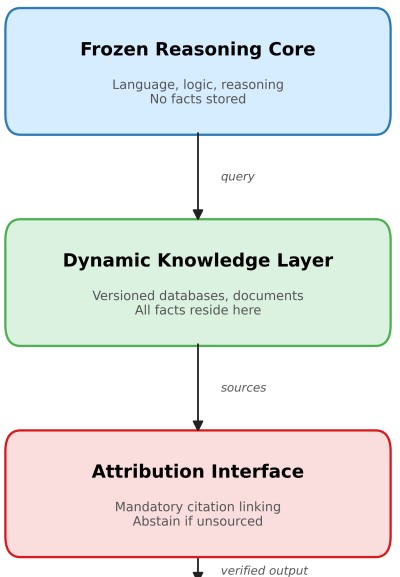

*Figure 2.* **Separation of Powers Architecture.** Unlike standard RAG which augments parametric knowledge, this architecture *prevents* parametric factual fallback for factual mode. The reasoning core handles linguistic competence; factual claims must be retrieved from a dynamic knowledge layer and pass attribution verification (or abstain).

RETRO (Borgeaud et al., 2022) achieves strong performance by externalizing knowledge; the Memorizing Transformer (Wu et al., 2022) shows external memory can match much larger models while enabling inference-time acquisition without gradient updates. An extended comparison table is in Appendix D.

## 5. Training Objectives and Evaluation

### 5.1. Predict-or-Retrieve Objective

Standard language modeling optimizes:

$$\mathcal{L}_{\text{LM}} = -\mathbb{E}_{(x,y)\sim\mathcal{D}}\left[\log P(y \mid x; \theta)\right],$$

which incentivizes completion even when evidence is absent. Referenceability requires different behavior: the system must (i) retrieve, (ii) condition generation on retrieved evidence, and (iii) abstain when evidence is insufficient.

We propose a **Predict-or-Retrieve** (PoR) objective that trains evidence-conditioned generation and calibrated refusal:

$$\mathcal{L}_{\text{PoR}} = -\mathbb{E}\left[\log P(y \mid x, r(x, \mathcal{K}); \theta)\right]$$
$$- \lambda \cdot \mathbb{E}\left[\text{reward}(\text{abstain} \mid \text{low-evidence})\right].$$

**Implementation implication.** For factual mode, the system should behave like a constrained generator rather than an omniscient oracle: it can only emit atomic claims it can attach to evidence; otherwise it must refuse or hedge. This shifts the core competency from "knowing facts" to *knowing how to ground facts*.

## 5.2. Training Against "Confident Bullshitters"

**Current RLHF can incentivize hallucination.** Human preference models often reward confident and complete answers over explicit uncertainty (Ouyang et al., 2022). In high-stakes settings, this incentive is misaligned: "I don't know" is sometimes the only safe output. Accordingly, factual-mode alignment should explicitly penalize unsourced confidence and citation fraud, while rewarding calibrated uncertainty and abstention.

Concretely, we advocate **inverting factual-task rewards**: penalize confident unsourced claims and citations that do not entail the associated claim; reward sourced correctness and honest refusal. A concrete scoring rubric is provided in Appendix C. This connects directly to selective prediction and calibrated refusal behavior: language models can learn calibrated uncertainty when such behavior is explicitly trained and rewarded (Kadavath et al., 2022; Kapoor et al., 2024).

## 5.3. Operationalized Evaluation Metrics

We propose three metrics that should be reported together:

**Attribution Density (AD):** fraction of atomic factual claims with valid citations.

$$\text{AD}(y) = \frac{|\{c_i \in y : \text{cited}(c_i) \wedge \text{valid}(c_i)\}|}{|\{c_i \in y : \text{factual}(c_i)\}|}.$$

AD must reflect *citation fidelity*, not citation count. We propose a tiered validation protocol for $\text{valid}(c_i)$ ranging from fast overlap checks to entailment and expert review (Appendix B).

**Parametric Leakage Ratio (PLR):** fraction of factual accuracy that persists when retrieval is disabled.

$$\text{PLR} = \frac{\text{Accuracy}(\text{retrieval=OFF})}{\text{Accuracy}(\text{retrieval=ON})}.$$

PLR is a counterfactual diagnostic: it measures whether the system can still "guess from weights." For sufficiently constrained high-stakes factual settings, lower PLR values are desirable because they indicate reduced dependence on unsupported parametric recall. A low PLR does not imply the absence of parametric world knowledge, but rather that factual answers are no longer *surfaceable* without evidence access. Importantly, PLR is intended as a diagnostic signal rather than a standalone optimization target. Extremely low

leakage may introduce trade-offs in latency, coverage, and usability, particularly in settings with incomplete or rapidly evolving evidence bases.

**Making leakage harder to dismiss (retrieval degradation curve).** To address the concern that "retrieval OFF" changes the interface distribution, we recommend reporting **retrieval degradation curves**:

- **OFF:** retrieval disabled.

- **SHUFFLE:** retrieval on, but passages shuffled across questions.

- **IRRELEVANT-$k$:** retrieval returns top-$k$ adversarially irrelevant passages.

- **REMOVE-GOLD:** retrieval on, but evidence-bearing passages removed (when gold is known).

A system that remains accurate under SHUFFLE/IRRELEVANT-$k$ is exhibiting parametric leakage (or verifier weakness). Reporting these conditions makes PLR substantially more robust than a single toggle.

**Abstention Accuracy:** ability to refuse when evidence is insufficient:

$$P(\text{abstain} \mid \text{insufficient evidence}).$$

This is the safety-critical complement to accuracy. It should be evaluated on datasets containing unanswerable questions (SQuAD 2.0 paradigm (Rajpurkar et al., 2018)) and reported as an abstention F1 that rewards both (i) high refusal under no-evidence and (ii) low refusal when strong evidence exists.

**Why these three metrics belong together.** Accuracy without AD measures fluency, not auditability. AD without leakage diagnostics can be gamed by retrofitting citations onto parametric guesses. Leakage diagnostics without abstention can still yield unsafe behavior if the model "tries anyway" when evidence is absent. Together, {AD, PLR/degradation, abstention} operationalize referenceability as a measurable system property rather than an aspiration.

Finally, systems should report these metrics as **curves**, not points: as attribution requirements tighten, what is the accuracy/coverage tradeoff? This motivates the referenceability curves in Figure 3 and reduces cherry-picked operating points.

# 6. High-Stakes Domains Validate the Position

## 6.1. Healthcare: Where Referenceability Is Mandatory

The feasibility of evidence-gated factual mode is reflected in the rise of citation-grounded healthcare systems and closed-

corpus clinical assistants (Appendix A summarizes representative designs and evidence with conservative language). In contrast, general-purpose LLMs in medical settings exhibit frequent citation failures and fabrication in multiple studies (Bhattacharyya et al., 2023; Gravel et al., 2023; Zhu et al., 2024), and deployed clinical ML systems have shown high miss rates in critical tasks (e.g., sepsis) (Goodman et al., 2024; Ross & Swetlitz, 2018). The lesson is not that medicine is "hard"; it is that **high-stakes deployment requires architectures where claims are inseparable from evidence**.

**Pre-emptive clarification.** We do not claim retrieval is error-free. We claim it is *auditable*: retrieval mistakes can be inspected, corrected, and versioned. Parametric mistakes typically cannot be traced to a source, making correction and compliance fundamentally harder. An auditable trail, at least in part, overcomes the challenges of attribution in legal disputes, such as in medical malpractice.

### 6.2. Legal and Scientific Domains

Legal practice has increasingly documented instances of fabricated or AI-hallucinated citations in court filings, leading to judicial sanctions, professional discipline, and evolving expectations that attorneys independently verify authority and factual support before submission (Charlotin, 2024; U.S. News & World Report, 2025). In scientific publishing and data infrastructure, failures of provenance manifest as poorly documented datasets, missing license information, and unverifiable claims, underscoring the need for systematic documentation such as datasheets to ensure transparency and accountability in research artifacts (Gebru et al., 2021). The principle generalizes: if referenceability and verifiable grounding fail in domains where truth is audited, the underlying architecture or documentation practice is inadequate for high-stakes use.

## 7. Addressing Key Objections

### 7.1. This Is Not Standard RAG

Standard RAG: "Use retrieval when available; fall back on parametric knowledge otherwise."

Our position: "parametric factual fallback is a liability". Current benchmarks often reward models for guessing correctly when retrieval fails. We argue this behavior is a safety hazard. We propose penalizing correct but unsourced guesses in high-stakes modes.

This changes: (i) **failure behavior** (abstention replaces guessing), (ii) **evaluation** (report AD/PLR/degradation/abstention, not only accuracy), and (iii) **training** (optimize epistemic behavior, not just fluency). The proposed contract (Section 4) is enforceable

and testable, making the delta operational rather than rhetorical.

### 7.2. Fluency and Mode Switching

We propose mode-aware constraints rather than a rigid binary separation. In high-stakes factual settings, factual spans should be evidence-gated, while non-factual or stylistic spans may remain unconstrained. In lower-stakes or creative settings, systems may permissibly rely more heavily on parametric generation. The active mode and its associated guarantees should remain transparent to the user. GopherCite (Menick et al., 2022) shows that fluency and citation can coexist when architecturally integrated.

### 7.3. Latency and Deployment Practicality

Retrieval introduces overhead, but this is an engineering trade-off rather than a conceptual blocker. Modern late-interaction retrieval systems such as ColBERTv2 demonstrate that neural retrieval can be both effective and efficient, achieving state-of-the-art search quality with substantially reduced index footprint and practical latency at scale; optimized engines such as PLAID further achieve query latencies on the order of tens of milliseconds (Santhanam et al., 2022b;a). Where retrieval is unavailable, systems should *degrade safely* by restricting factual coverage, clearly labeling uncertainty, and abstaining on high-risk queries. The core claim remains: absent access to verifiable evidence, factual-mode generation must not impersonate grounded knowledge. The trade-off therefore shifts from maximizing unconstrained fluency toward explicitly balancing safety, coverage, latency, and auditability.

## 8. Call to Action

**Stop benchmarking accuracy without provenance.** Reporting "92% on Natural Questions" without attribution is not a safety signal; it is a fluency signal. For factual-mode systems, progress must be reported as a *bundle*: predictive accuracy *and* provenance metrics. Concretely, papers should report (i) **Attribution Density (AD)** to quantify claim-level citation coverage, (ii) **Parametric Leakage Ratio (PLR)** (with degradation controls) to expose dependence on retrieval versus weights, and (iii) **Abstention Accuracy** to measure whether the system refuses when evidence is insufficient. A model with slightly lower raw accuracy but high AD and high abstention accuracy is often *more deployable* than a higher-accuracy model whose claims cannot be audited.

**Stop treating fine-tuning as knowledge injection.** In a synthetic-data regime, "injecting facts into weights" is not merely an engineering shortcut; it is a provenance-destroying operation. Knowledge injection papers should

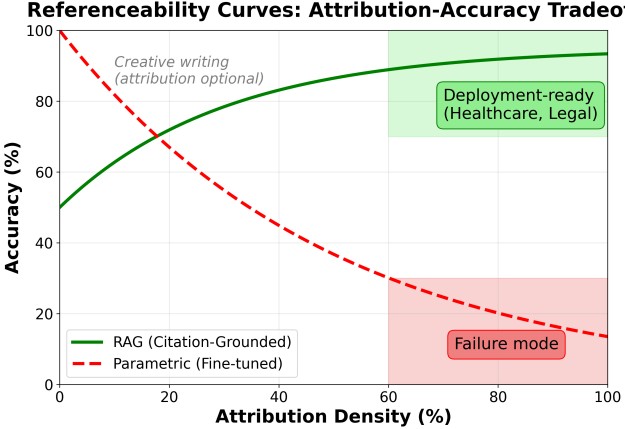

*Figure 3.* **Referenceability Curves.** As attribution requirements increase (x-axis), accuracy may initially decrease (y-axis). Systems that maintain high accuracy with high attribution (top-right) are superior for high-stakes deployment. Purely parametric systems cannot achieve meaningful attribution without evidence access.

therefore be held to reporting standards that reflect real deployment constraints: update cost and latency of updates, verification mechanisms for injected knowledge, long-tail robustness, and compliance properties (including deletion/unlearning implications) (Ji et al., 2023). If a method cannot state *where a fact came from* after it is injected, it has not solved grounding—it has only hidden knowledge inside an unauditable substrate.

We acknowledge that scientific knowledge will be updated continuously, and hence atomic facts may evolve into 'general knowledge' with time. Benchmarks will need to be updated to reflect such evolution.

**Publish referenceability curves, not single numbers.** The community needs an analogue of ROC curves for grounded generation, plotting attribution requirements against accuracy or coverage (Figure 3). These curves make deployment suitability explicit: systems that sustain accuracy under high attribution density are deployment-ready, while those that degrade sharply are unsuitable for high-stakes factual use. Single operating points obscure this trade-off and invite cherry-picking.

**Elevate abstention as a first-class success case.** Benchmarks should include unanswerable or insufficient-evidence questions and reward appropriate refusal. "I cannot answer this without evidence" should be scored as a success, not a failure. Without abstention evaluation, systems are implicitly trained to attempt every question—the exact behavior that converts uncertainty into confident fabrication.

## 9. Alternative Views

We acknowledge credible counterarguments and clarify that our claims are scoped to high-stakes factual mode under synthetic-data contamination and compliance constraints.

**View 1: "Parametric knowledge is a capability—removing it wastes useful signal."** We agree that pretraining confers broad linguistic competence and world knowledge, and that retrieval-augmented systems can effectively combine parametric and non-parametric memory to improve factuality over parametric-only generation (Lewis et al., 2020; Chen et al., 2025). Recent work further explores hybrid designs that integrate retrieved evidence with parametric verification and adaptive retrieval strategies (Su et al., 2025). Our claim is narrower: in high-stakes settings, factual *outputs* should be evidence-conditioned rather than justified solely by parametric recall. Referenceability constrains what is *surfaced*, not what is encoded in parameters.

**View 2: "Hybrid is enough: use retrieval when available, fall back on the model otherwise."** Hybrid pipelines are common, but their default fallback behavior is precisely the epistemic failure mode at issue (Lewis et al., 2020; Rayo et al., 2025; Kozłowski, 2023). If the system can answer factual questions with retrieval disabled, users cannot distinguish sourced statements from unsourced ones, and the architecture remains vulnerable to synthetic contamination, staleness, and unverifiable error (Kapoor et al., 2024). This motivates reporting leakage diagnostics; for high-stakes factual mode, lower leakage becomes increasingly desirable as deployment criticality increases.

**View 3: "Closed corpora limit coverage and prevent synthesis across sources."** Coverage limits are real, but the appropriate response under limited evidence is calibrated uncertainty and abstention, not hallucinated completeness (Wen et al., 2025; Savage et al., 2025; Tao et al., 2025). Evidence-based synthesis remains possible by retrieving across multiple sources, representing disagreement, and surfacing contradictions when they exist. What must be rejected is synthesis driven primarily by parametric recall when retrieval is weak.

**View 4: "Referenceability is too expensive to evaluate at scale."** If evaluation cannot validate claim–citation fidelity even approximately, then the system is not being measured for the property it claims to provide (Mugaanyi et al., 2024). Practical compromises exist: tiered validation using fast overlap checks, entailment verification, and periodic expert audits yields tractable signals while reserving deep review for high-risk outputs (Tang et al., 2025; Liu et al., 2025). This tiered approach acknowledges that while automated verification is not perfect, it provides an audit trail that

purely parametric systems entirely lack.

**View 5: "Perfection is unrealistic; domains tolerate approximate answers."** Perfection is neither realistic nor required, but epistemic communication is. Models should preserve and surface genuine uncertainty rather than collapse ambiguity into confident outputs; soft-label training preserves epistemic uncertainty by aligning model confidence with human disagreement (Singh et al., 2025). Current interfaces lack structured mechanisms for specifying epistemic preferences, leading to ad-hoc prompting rather than reliable control (Clark et al., 2025). In high-stakes settings, systems must trace claims to evidence and signal uncertainty to remain auditable (Warren et al., 2025); accordingly, systems may provide partial evidence, competing evidence, or refusal, but must not present unsourced claims as auditable truth.

**Summary.** These views may suffice for creative or low-stakes use, but fail in high-stakes factual settings where fluent outputs lack recoverable lineage. We therefore argue for an architectural, measurable response: evidence-first generation, training for groundedness or abstention, and reporting provenance metrics alongside accuracy.

## 10. Operational Challenges and Open Research Questions

Several important challenges remain unresolved.

**Contested and evolving evidence.** High-stakes domains frequently contain conflicting guidelines, incomplete evidence, or temporally evolving recommendations. Referenceability does not eliminate these disagreements; rather, it externalizes them into inspectable evidence traces.

**Multi-hop reasoning and joint entailment.** Many factual claims require synthesis across multiple retrieved sources. While existing retrieval and verification systems increasingly support multi-document reasoning, robust joint-entailment verification remains an open challenge.

**Verifier and retrieval reliability.** The proposed framework depends on retrieval quality and verifier calibration, both of which introduce their own failure modes. Incorrect retrieval, non-entailing citations, and verifier errors may still produce misleading outputs despite explicit attribution.

**Coverage versus abstention.** Evidence-gated systems may refuse more frequently in settings where evidence is sparse, incomplete, or rapidly changing. This introduces an explicit trade-off between safety, usability, and coverage.

**Referenceability is not equivalent to truth.** A system may be perfectly referenceable yet still grounded in incomplete, biased, or incorrect evidence. Our proposal therefore focuses on auditability and epistemic transparency rather than guarantees of factual correctness.

## 11. Conclusion

Synthetic training regimes undermine the epistemic assumptions that once made parametric knowledge appear grounded, revealing failures in factual reliability, unlearning, and auditability. As a result, referenceability must be treated as a first-class architectural requirement rather than an optional retrieval add-on. We argue for a strict provenance–parametric separation: parametric models for language and reasoning, non-parametric mechanisms for factual grounding, and mandatory attribution or abstention.

Importantly, we do not claim that evidence-gated architectures eliminate factual error or guarantee truth. Rather, they transform factual generation into a process that is externally inspectable, contestable, and auditable. In high-stakes settings, where claims must remain temporally accountable and evidentially traceable, provenance-aware architectures are likely to become a practical requirement rather than an optional design preference.

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

## A. Healthcare AI Systems: Architectural Patterns and Evidence

This appendix provides *conservative* summaries of representative citation-grounded clinical AI designs. We intentionally avoid unverifiable marketing claims (e.g., valuations) and clearly distinguish: (i) independently evaluated findings (when available) versus (ii) vendor-described architecture. The goal is not endorsement, but illustrating that evidence-gated designs are implementable and increasingly demanded by workflow and regulation.

These examples are not offered as evidence of superiority, but as proof that evidence-gated architectures are technically and operationally feasible today.

### A.1. Example Pattern: Closed-Corpus Retrieval with Inline Claim-Level Citations

Several clinical assistants are described as operating primarily over curated medical corpora (e.g., guidelines, peer-reviewed literature, licensed reference content), producing responses with inline citations. Architecturally, these systems instantiate the "Separation of Powers" pattern:

- retrieval over a bounded corpus with provenance metadata,

- synthesis constrained by retrieved passages,

- UI and logging that expose claim–source links.

When independent evaluations exist, they typically measure clinical correctness *and* citation fidelity (or at minimum citation relevance). For example, some retrospective or user studies report that clinicians value source-linked outputs and that citation support improves trust calibration (**?**Robleto et al., 2024). We treat such findings as suggestive rather than universal guarantees.

## A.2. OpenEvidence (illustrative; architecture described publicly)

Public descriptions characterize OpenEvidence as a medical AI platform emphasizing retrieval over medical literature and returning inline citations to supporting sources (**?**). The key architectural relevance to our position is not brand-specific performance, but the evidence-gated interaction pattern: citations are presented as audit hooks and uncertainty is surfaced when evidence is weak or conflicting (as reported in available evaluations/documentation).

### A.3. Clinical reference assistants built on licensed medical content

Clinical assistants built on licensed medical reference corpora (e.g., textbooks, journals, proprietary clinical summaries) similarly operationalize evidence gating by restricting generation to a bounded, curated knowledge base and linking outputs to the underlying content (**?**). Again, the salient point is the architecture: *bounded corpora + claim-linked sources*, not the specific vendor.

### A.4. Comparative framing: general LLMs vs evidence-gated designs

Multiple studies have documented citation errors or fabricated references in general-purpose LLM outputs in biomedical contexts (**?**Gravel et al., 2023; Zhu et al., 2024). Evidence-gated designs do not eliminate error, but they change the failure mode from "unverifiable" to "inspectable," enabling correction and safer integration into clinical workflows.

## B. Attribution Validation Protocols and Reproducible Scoring

This appendix operationalizes valid($c_i$) for Attribution Density and specifies a reproducible evaluation unit.

### B.1. Unit of Evaluation: Claim–Citation Pairs

A system output is decomposed into **atomic factual claims** $\{c_i\}$. Each factual claim must be associated with one or more evidence identifiers $\{e_{ij}\}$ pointing to a passage span in a source document (with metadata: doc id, section/span offsets, timestamp).

The fundamental evaluation object is a **claim–citation pair** $(c_i, e_{ij})$.

### B.2. Claim Atomization Procedure (Practical Recipe)

We recommend a constrained recipe that is reproducible and robust:

1. **Sentence segmentation:** split into sentences; further split on conjunction patterns that typically join independent propositions (e.g., "and", "but", "however", "whereas") using a conservative rule set.

2. **Factuality filter:** classify each segment as factual vs non-factual using either: (i) a lightweight classifier trained on examples (preferred), or (ii) an LLM forced into a binary label with exemplars and a strict output schema.

3. **Normalization:** convert each factual segment into a proposition form (optional but useful for entailment).

4. **Evidence binding:** require that the generator emit evidence IDs inline (or as structured metadata) so mapping is explicit, not inferred.

### B.3. Three-Tier Validation Stack for valid($c_i$)

**Tier 1: Lexical/Semantic Overlap (fast, high recall).** Compute ROUGE-L and/or BERTScore between the claim and cited passage. Use a conservative threshold to flag potential support; do not treat Tier 1 as sufficient for safety claims.

**Tier 2: NLI Entailment (medium speed, higher precision).** Apply an entailment model (e.g., DeBERTa-v3 (He et al., 2021)) to classify whether the cited passage entails the claim. Accept only "entailment"; reject "neutral" and "contradiction."

**Tier 3: Expert Review (slow, gold standard).** For medical/legal deployments, sample claim–citation pairs for domain-expert verification, using Tier 3 labels to calibrate Tier 1/2 thresholds and monitor drift.

**Reporting.** Systems should report AD at each tier (AD@1, AD@2, AD@3) to separate UI-level citation coverage from true evidential support.

## C. Example Factual-Mode Reward Rubric

A concrete (illustrative) rubric for factual-mode reward modeling:

- **+2:** Correct answer with valid source citation (claim-level support).

- **+1:** Calibrated uncertainty with partial evidence (states limits and cites what exists).

- **0:** Abstention when evidence is unavailable (states what is needed to answer).

- **-2:** Incorrect answer with citation (citation does not support the claim; "citation fraud").

- **-5:** Confident unsourced factual claim (hallucination in factual mode).

This aligns optimization with epistemic integrity: in factual mode, refusal is preferable to unverifiable completion.

## D. Architectural Comparison (Extended)

## E. Agentic LLMs with Tool Calling: Citation Architecture Analysis

Major LLM providers have implemented citation capabilities through tool calling and web search integration, demonstrating feasibility while revealing that *optional* citations preserve parametric fallback failure modes.

### E.1. OpenAI Responses API: Structured Citation Support

OpenAI's Responses API provides citation annotations when web search or document grounding is enabled (OpenAI, 2025). The system can return structured citation metadata mapping text segments to sources, enabling automated validation. However, enforcement depends on application configuration; citations are not necessarily mandatory for every factual span in general deployments (Wu et al., 2025).

### E.2. Anthropic Citations API: Document-Based Attribution

Anthropic's Citations API enables citations over provided documents and reports reductions in source hallucinations in customer workflows (Anthropic, 2025). As with other providers, enforcement depends on application-level constraints.

### E.3. Google Gemini: Search Grounding and Verification

Google Gemini provides grounding metadata for some responses (Google AI, 2025). Retroactive verification features can still risk "citation shopping" if generation precedes retrieval, motivating evidence-first ordering.

### E.4. Implications for Our Position

Current systems validate technical feasibility; the gap is architectural enforcement: moving from "citations available" to "citations required in factual mode."

## F. Regulatory Landscape: Citation Requirements in High-Stakes AI

### F.1. FDA Guidance: Clinical Decision Support Software

The 21st Century Cures Act's Criterion 4 requires that CDS software enable clinicians to **"independently review the ba-**

**sis for such recommendations."** FDA guidance interprets this as providing sufficient information for review including peer-reviewed studies and guidelines. Citation-grounded architectures align with this requirement; purely parametric systems cannot expose a reviewable basis.

### F.2. EU AI Act: Transparency for High-Risk Systems

EU Regulation 2024/1689 (AI Act, applicable August 2026) classifies many medical AI systems as high-risk and requires sufficient transparency for deployers to interpret outputs. Citation-grounded systems provide a practical transparency substrate via sources.

## G. Implementation Guidelines for Healthcare AI

### G.1. Building Retrieval-Augmented Medical AI

**Knowledge base construction:**

1. Index peer-reviewed medical literature (PubMed, guidelines)

2. Add temporal metadata (publication date, update timestamp)

3. Include evidence-strength heuristics (systematic reviews > RCTs > case studies)

4. Implement contradiction detection for conflicting sources

**Retrieval mechanism:**

1. Dense retrieval using domain embeddings (BioBERT, PubMedBERT)

2. Rerank by recency, evidence quality, and relevance

3. Return top-$k$ with provenance (DOI, authors, publication)

**Generation with attribution:**

1. Condition generation on retrieved context

2. Require explicit citation for each factual claim

3. Implement abstention threshold based on retrieval confidence

4. Log all retrievals for audit trail

**Validation:**

1. Domain expert review of claim-citation pairs

*Table 3.* **Architectural Comparison: Parametric vs. Separation of Powers**

| Capability | Parametric (Fine-Tuning) | Separation of Powers (Evidence-First) |
| --- | --- | --- |
| Knowledge updates | Retraining or fragile edits; slow | Index updates; fast and auditable |
| Attribution | No native source lineage | Claim-to-source linking by design |
| Temporal validity | No staleness mechanism | Versioning enables "as-of" queries |
| Long-tail facts | Requires extreme scaling (Kandpal et al., 2023) | Retrieved directly from corpora |
| Audit trail | None (weights opaque) | Logged retrieval + citation trails |
| Hallucination handling | Must output tokens | Abstain when evidence insufficient |
| Legal compliance | Unlearning intractable (Bourtoule et al., 2021) | Deletion and verification feasible |

2. Long-tail evaluation (rare conditions)

3. Stress test with contradictory sources

4. Measure AD, leakage diagnostics, and abstention accuracy

