# OpenReview forum: "Position: The Data Provenance–Parametric Divide in Large Language Models"
_ICML.cc/2026/Position_Paper_Track — ICML 2026 Position Paper Track regular_

### Official Review · Reviewer_nS5H · 2026-03-12

**Significance:** 4
**Argument Clarity:** 3
**Rating:** 2
**Confidence:** 4

**Questions:**

1. Can the authors provide concrete estimates (even order-of-magnitude) for the performance degradation expected when enforcing PLR ≈ 0 on existing attributed QA benchmarks?
2. How should questions like "are all swans white" be handled? Suppose all your retrieved facts indicate that swans are white. Would it be more appropriate to answer yes or no?
3. The paper proposes that claim atomization can be done with a "lightweight classifier." Has this been tested, and what are realistic precision/recall numbers for factual vs. non-factual classification on clinical text?

**Alternative Views Section:**

Yes

**Compliance With Llm Reviewing Policy A Conservative:**

Affirmed.

**Discussion Potential:**

3

**Final Justification:**

Authors' rebuttals partially address my concerns. My main concern related to the limitation of the work still remains, and I decide to keep my score.

**Paper Summary:**

This position paper argues that as LLMs increasingly train on synthetic data, the implicit assumption that "if it's in the weights, it came from somewhere real" is no longer valid. The authors identify an epistemic gap: parametric models cannot distinguish knowledge learned from genuine evidence versus inherited from synthetic generation, creating "data laundering" where errors gain legitimacy through repetition. The paper proposes a strict "provenance–parametric separation": parametric models should handle language and reasoning, while factual claims in high-stakes settings must be grounded in non-parametric, retrievable evidence. The core technical contribution is a "Factual Mode" invariant: no atomic factual claim may be emitted unless attached to evidence identifiers whose cited passages entail the claim; otherwise, the system must abstain. This is operationalized through a "Separation of Powers" three-tier architecture (frozen reasoning core, dynamic knowledge layer, mandatory attribution interface), a Predict-or-Retrieve training objective, and three evaluation metrics. The position is applied to healthcare, legal, and scientific domains, and the authors provide a detailed call to action for the research community.

**Position:**

Yes

**Position In Title:**

Yes

**Related Work:**

3

**Strengths And Weaknesses:**

## Strengths

- The concern about synthetic data contamination undermining epistemic trust in LLM outputs is well-founded and increasingly urgent.
- The three proposed metrics—AD, PLR, and Abstention Accuracy—are well-defined and actionable. Reporting referenceability curves rather than single operating points is a valuable methodological contribution.

## Weaknesses

- It's unclear whether current architectures can achieve near-zero PLR while maintaining useful performance. The paper would benefit from a deeper discussion of the performance–referenceability tradeoff curve for existing systems.
- The paper correctly identifies this as "the hardest operational question," but the proposed solution—rule-based segmentation plus a lightweight classifier—may fall short for complex multi-sentence claims common in clinical and legal settings. The gap between aspiration and practical difficulty deserves more attention.
- Real-world systems struggle to cleanly separate parametric reasoning from factual retrieval. For instance, the "frozen reasoning core" still encodes world knowledge that influences reasoning. How to prevent this knowledge from leaking into factual claims remains unclear.
- The paper proposes a solution to a well-known LLM problem: reducing hallucinations. With experimental data, it might be better suited for the main track than as a position paper.

**Support:**

2

---

> ### Author Rebuttal · Authors · 2026-03-30
>
> We thank the reviewer for the careful and critical evaluation of our work. We believe that several of the concerns raised, particularly regarding feasibility, lack of empirical validation, and scope, reflect a difference in expectations between Position Track contributions and System Track criteria, and we respectfully clarify this distinction.
>
> This submission is intentionally framed as a *Position Paper*, whose goal is not to present a benchmark-optimised implementation, but to define normative safety requirements for large language models in high-stakes settings. Specifically, we introduce the concept of *referenceability* as a system invariant, together with measurable constructs such as *Attribution Density (AD)*, *Parametric Leakage Ratio (PLR)*, and *Abstention Accuracy*. Our argument is that systems which do not satisfy these properties remain fundamentally non-auditable, regardless of their empirical accuracy.
>
> On the suggestion that this work would be more appropriate for the Main Track, we respectfully disagree. Requiring empirical validation as a precondition for articulating such constraints creates a circular dependency, where safety principles can only be introduced after they are already engineered and validated. In contrast, the Position Track exists precisely to surface such system-level requirements before full realisation. We therefore view the absence of empirical benchmarking not as a limitation, but as a deliberate alignment with the track’s intent.
>
> Regarding feasibility and the *Parametric Leakage Ratio (PLR)*, we agree that enforcing near-zero leakage is challenging and may reduce coverage. However, our proposal does not assume that PLR can or should be minimised to zero. Rather, PLR is introduced as a diagnostic construct, enabling explicit measurement of when unsupported parametric knowledge is surfaced. Recent work on memorisation and benchmark leakage (e.g., Carlini et al., 2023; Kandpal et al., 2023; Sainz et al., 2024) suggests that such leakage is both measurable and increasingly unavoidable in large-scale models. Our contribution is to elevate this into a system-level accountability signal, making the safety–coverage trade-off explicit. In high-stakes contexts, abstention is not a failure mode, but a desirable safety outcome.
>
> On the role of parametric knowledge, we fully agree that the reasoning core encodes substantial world knowledge. Our proposal does not attempt to eliminate this. Instead, the constraint is behavioural rather than representational: in *Factual Mode*, the model is not permitted to surface claims as authoritative outputs unless they are supported by retrievable evidence. This aligns with emerging paradigms that decouple reasoning from knowledge (e.g., Lewis et al., 2020; Borgeaud et al., 2022), treating LLMs as reasoning engines rather than knowledge stores. Leakage from the reasoning core cannot be eliminated at the representation level; instead, the proposal enforces output-level constraints on what may be presented as factual.
>
> On atomisation, we acknowledge that reliable claim decomposition is challenging, particularly for complex, multi-sentence reasoning. However, this is not purely speculative. Prior work such as *FactScore* (Min et al., 2023) and subsequent claim verification pipelines (e.g., ClaimVer, 2024–2025) demonstrate that atomic fact decomposition can be operationalised for long-form generation. Our position does not depend on perfect atomisation; rather, even approximate decomposition provides a meaningful unit of audit, which is strictly safer than monolithic, unverifiable outputs.
>
> Finally, the “all swans are white” example illustrates the failure mode we aim to address—*epistemic provenance collapse*, where models hallucinate consensus from parametric memory. Under our proposed constraint, the system would instead qualify or abstain based on retrieved evidence, aligning outputs with a traceable evidential snapshot.
>
> We hope these clarifications better position the work within the intended scope of the Position Track, and we kindly ask the reviewer to reconsider their assessment in light of this framing.
>
> **References**
>
> **[Doshi-Velez17]** Towards a Rigorous Science of Interpretable Machine Learning. *arXiv*
> **[Thorne18]** FEVER: A Large-Scale Dataset for Fact Extraction and Verification. *NAACL*
> **[Lewis20]** Retrieval-Augmented Generation for Knowledge-Intensive NLP. *NeurIPS*
> **[Sendak20]** A Path to Real-World Clinical AI. *Nature Medicine*
> **[Borgeaud22]** Improving Language Models by Retrieving from Trillions of Tokens. *ICML*
> **[Kandpal23]** Large Language Models Struggle to Learn Long-Tail Knowledge. *ICML*
> **[Carlini23]** Extracting Training Data from Large Language Models. *USENIX Security*
> **[Jiang23]** Active Retrieval Augmented Generation. *EMNLP*
> **[Min23]** FactScore: Fine-grained Atomic Evaluation of Factual Precision. *EMNLP*
> **[Sainz24]** NLP Evaluation in the Age of Large Language Models. *arXiv*

---

> > ### Author Rebuttal · Reviewer_nS5H · 2026-04-03
> >
> > Thank the authors for the rebuttal, which partially resolves my concerns. However, I am still confused about the argument related to “all swans are white.” Should the system always output “don’t know,” or what kinds of atomic facts are sufficient to support the claim in the current setting?

---

### Official Review · Reviewer_KGFh · 2026-03-12

**Significance:** 3
**Argument Clarity:** 2
**Rating:** 3
**Confidence:** 3

**Questions:**

+ How exactly do you split answers into checkable factual units, and do you have evidence that your automatic splitting + entailment checking is reliable enough to be used in real auditing?

+ What safeguards ensure the evidence itself is high-quality and resistant to attacks, so that a high citation rate (high AD) doesn’t mislead people into thinking the answer is safe when the sources are bad?

+ How can you combine your evidence-gating design with modern retrieval-learning and refusal-calibration methods so the system not only cites evidence and also can provide guarantees about when it answers and refuses?

+ Your system says each claim must be ‘entailed by evidence.’ But many real answers require combining facts from multiple sources. Do you allow that, and if your answer is yes, how do you check that multiple passages jointly support the claim?

+ What could be your threat model, and how will you prevent the system from hoosing easy or low-quality sources? How will you handle verifier/NLI errors, and will you share your verifier calibration and evaluation procedure?

**Alternative Views Section:**

Yes

**Compliance With Llm Reviewing Policy A Conservative:**

Affirmed.

**Discussion Potential:**

3

**Final Justification:**

The main strengths are that the paper presents a clear idea, proposes useful evaluation metrics, and raises an important question about factual grounding and provenance. The provided rebuttal by the authors partially addresses my concerns. However, some parts are still ambiguous, and my main concern is whether this framework is reliable enough in practice, and the rebuttal did not fully provide concrete evidence or detailed methods for that, so I will maintain my current evaluation.

**Paper Summary:**

This position paper argues that the parametric LLMs cannot function as trustworthy witnesses of factual provenance in an era of growing synthetic data. The authors say that enforcing referenceability as a system for high-stakes factual use which every claim must be evidence-gated by explicit source identifiers or the system must abstain. The authors propose a architecture, "separation-of-powers" that decouples a frozen reasoning core from a dynamic, versioned knowledge layer and a mandatory attribution interface. They also provide evaluation metrics.

**Position:**

Yes

**Position In Title:**

Yes

**Related Work:**

2

**Strengths And Weaknesses:**

Strengths

+ This paper describes, a clear, enforceable design principle, which is no factual output without evidence.

+ The authors propose measurable, testable construct (AD, PLR, abstention metrics), and these has the possibility to be adopted by the community to quantify provenance, leakage, and refusal behavior.

+ The authors give a practical guidelines for building an LLM system that reasons with a model, but sources facts from an auditable knowledge base.

+ The paper is clearly written, it defines key concepts, explains how systems fail, and clearly shows how its approach differs from common RAG/citation methods.

Weaknesses

+ The paper talks about "atomic factual claims", but it doesn't fully explain how to reliably split answer into those tiny facts in a way that works automatically, because it can be still unreliable, and errors can make the system either cite too much or fail to cite when it should.

+ PLR is interesting, however if you require it to be almost zero in every ‘factual mode’ use case, you might force the systems to depend on the retrieval even when it's unnecessary or impossible, which could make them less practical.

+ Regarding the PoR, the authors only describes it conceptually, it doesn't provide enough concrete training details or robustness analysis to show it can work reliably in real settings.

+ Another weakness can be that the architecture is only as safe as the retrieval and verification components. However the paper doesn't fully address how to handle retrieval mistakes, wrong sources, or verifier weaknesses, so the system could still produce incorrect answers that look trustworthy.

+ There is no latency/cost analysis for the evidence-gated stack in realistic deployments.


+ The authors talk about abstention, however they don't use or discuss established methods that can formally guarantee how safe the answer or refuse decision is, so the abstention part remains heuristic rather than guaranteed.

**Support:**

3

---

> ### Author Rebuttal · Authors · 2026-03-30
>
> We thank the reviewer for the detailed and thoughtful feedback, particularly regarding feasibility, atomisation, and system robustness.
>
> On claim atomisation, we agree that reliable decomposition is challenging, particularly for complex reasoning. However, our proposal does not depend on perfect atomisation. Rather, it introduces a practical granularity of audit, where even approximate decomposition enables partial verification of outputs. Prior work has demonstrated that fine-grained factual evaluation through atomic claims is feasible in practice [Min23]. This allows verification to operate at a more meaningful level than treating outputs as monolithic units.
>
> On multi-hop reasoning, we agree that many claims require synthesis across multiple sources. The proposed framework accommodates this through set-based evidence attribution, where claims are supported collectively by multiple pieces of evidence. This is consistent with prior work on multi-evidence verification and retrieval-based reasoning [Thorne18; Jiang23].
>
> On PLR and abstention, we agree that enforcing low leakage introduces a trade-off with coverage. This trade-off is central to our argument. Evidence of memorisation and uneven knowledge distribution in LLMs [Carlini23; Kandpal23], as well as concerns around benchmark contamination [Sainz24], further motivate the need to explicitly track when models rely on unsupported parametric recall. PLR is therefore introduced as a diagnostic signal, not an optimisation objective.
>
> On the Predict-or-Retrieve objective, we acknowledge that it is presented conceptually. This is intentional and consistent with the Position Track. The contribution is to define a behavioural contract, rather than a specific optimisation pipeline.
>
> On retrieval and verifier reliability, we agree that these components introduce their own failure modes. Our claim is not that such errors are eliminated, but that they become observable and auditable, aligning with broader goals of interpretable and accountable AI [Doshi-Velez17].
>
> We appreciate the reviewer’s feedback and will incorporate these clarifications.

---

> > ### Author Rebuttal · Reviewer_KGFh · 2026-04-03
> >
> > Thank you for the rebuttal. The rebuttal partially addresses the conceptual questions around approximate atomization, multi-source attribution, and the role of PLR/abstention. However, it still does not fully answer my main concerns about robustness. In particular, it does not show whether automatic claim splitting and entailment checking are reliable enough for real audits, how the system makes sure the sources themselves are trustworthy, how it checks claims that need support from several sources together, what threat model it assumes, or how the verifier is calibrated and evaluated. So overall, the rebuttal helps explain the idea better, but it does not fully address my concerns about whether this framework is strong enough to support serious safety or auditing claims in practice.

---

### Official Review · Reviewer_YaeV · 2026-03-12

**Significance:** 2
**Argument Clarity:** 3
**Ethics Flag:** Yes
**Rating:** 4
**Confidence:** 3

**Questions:**

Please see weakness

**Alternative Views Section:**

Yes

**Compliance With Llm Reviewing Policy A Conservative:**

Affirmed.

**Discussion Potential:**

3

**Final Justification:**

My concerns have been resolved. I will maintain my score.

**Paper Summary:**

This work studys the key challenge of how to enforce verifiable data provenance and explicits claim-to-evidence attribution as non-negotiable system invariants to resolve the epistemic crisis caused by the blending of parametric knowledge and synthetic data in LLMs.

**Position:**

Yes

**Position In Title:**

Yes

**Related Work:**

3

**Strengths And Weaknesses:**

Strengths

1. This paper study how enforcing mandatory claim-to-source linking can transform black-box LLMs into auditable systems, ensuring every factual output is backed by verifiable evidence.

2. They studied the key challenge of decoupling reasoning from knowledge storage, which effectively prevents the accumulation of errors and "data bleaching" caused by models training on their own synthetic outputs.

Weaknesses

1. The problem addressed is significant, but the practical feasibility remains low. From a technical standpoint, all data—regardless of its source or factual accuracy—plays a crucial role in the model's pre-training phase. Furthermore, accurately labeling massive datasets is an inherently daunting task. Even if such labeling were possible, there remains the critical question of who would verify and guarantee the correctness of these annotations.

2. The authors focus primarily on the inherent flaws of models, yet the models themselves are highly malleable and offer significant room for architectural modification. The core challenge lies in how to effectively distinguish between different types of data (e.g., synthetic vs. human-authored). Does the paper propose any robust methodology or "best practices" to address this fundamental classification problem?

3. How can service providers be incentivized to adopt a framework that likely compromises model performance and user experience? Alternatively, is there a technical pathway to implement these provenance requirements without sacrificing the efficiency and fluency that current users expect?

**Support:**

3

---

> ### Author Rebuttal · Authors · 2026-03-30
>
> We thank the reviewer for the positive and encouraging feedback, and for recognising the importance of auditable, evidence-backed LLM systems.
>
> We clarify that our proposal does not require large-scale reconstruction of dataset-level provenance. Instead, provenance is enforced at runtime, where claims surfaced in Factual Mode must be supported by retrievable, versioned evidence. This avoids the need to infer provenance from model parameters and instead externalises knowledge into an inspectable evidence layer, consistent with retrieval-augmented paradigms [Lewis20; Borgeaud22].
>
> We agree that LLMs are inherently adaptable, and our proposal can be understood as an architectural adaptation to this reality. By separating generation from factual authorisation, the system ensures that outputs presented as factual remain externally verifiable and auditable, even as the underlying model evolves.
>
> We also agree that incentives for adoption are critical. In high-stakes domains such as healthcare, deployment decisions are often governed by requirements for auditability, traceability, and performance monitoring [Sendak20]. Systems that provide explicit evidential grounding are therefore better aligned with real-world deployment constraints.
>
> We further emphasise that this approach reframes trade-offs in terms of safety versus coverage, rather than accuracy alone. In scenarios where evidence is unavailable or insufficient, abstention is not a failure but a desirable safety behaviour.
>
> We appreciate the reviewer’s supportive evaluation and will refine the manuscript to further clarify these points.

---

### Official Review · Reviewer_vVGp · 2026-03-13

**Significance:** 3
**Argument Clarity:** 2
**Rating:** 3
**Confidence:** 4

**Questions:**

Questions
- How would the separation-of-powers architecture handle claims that require multi-hop reasoning across several retrieved documents? The current description seems to assume a relatively simple retrieve-then-generate pipeline, but clinical reasoning often requires synthesizing information from multiple, potentially contradictory sources.
- Is there reliable quantitative measurement available to confirm the assertion that zero PLR values are achievable without a major loss in coverage? Have you done your own verification of the current RAG systems?
- If there is limited quantity or quality of evidence supporting clinical guidance (e.g., rare diseases), what effect will that have on the credibility or reliability of the guidance generated from this evidence? Although the 3-tier validation system assures citations are accurate, it does little to provide trustworthiness of the actual evidence itself. Is there a concern that, if a decision is made based on false representations of accuracy, there can be false impressions given to the system users and providers?
- Since the atomization procedure depends on accurate classification of claims as factual or non-factual, what happens when the system classifies a factual claim incorrectly? If a claim is incorrectly identified as non-factual, the system will not process that fact and therefore cannot rely on that fact as evidence and thus affect the overall accuracy of the system. To ensure the integrity of the system and provide trustworthiness to all users of the system, how do you see the system achieving the needed level of reliability?
- How do you see existing LLM systems transitioning to the proposed system? You indicate that the proposed architecture is a clean-slate approach but in reality most systems in use today were developed with legacy technologies and thus cannot be easily transitioned to new architectures without significant investment on the part of both end users and the vendors implementing the new technologies?

**Alternative Views Section:**

Yes

**Compliance With Llm Reviewing Policy A Conservative:**

Affirmed.

**Discussion Potential:**

2

**Final Justification:**

The authors' explanations solve my concerns regarding the separation-of-powers framework, retrieval grounding, and factual mode behavior. I encourage you to incorporate these in the revision for better persuasiveness. Nevertheless, some questions remain regarding conceptual precision, e.g., how does this framework handle contested evidence, and how can PLR evaluation avoid confounding? Therefore, I maintain my score.

**Paper Summary:**

This position paper argues that the increase in large language model use of synthetic data negates the reliability of their parametric representations. They argue that referenceability must be included as a non-negotiable invariant for high-stakes systems; therefore, no factual claims will be emitted from the system without evidence and will be prevented from doing so otherwise.

**Position:**

Yes

**Position In Title:**

No

**Related Work:**

2

**Strengths And Weaknesses:**

Strengths
- The paper identifies a legitimate fail condition that has not been sufficiently explored; the loss of epistemic provenance due to the emergence of synthetic data. The dynamic of a model generating output based on another model’s output, which creates a “citation ouroboros," is well articulated and will become increasingly relevant and problematic as synthetic data becomes a more significant proportion of all data used for training.
- The authors create a clear and purposeful distinction between using retrieval as augmentation in traditional retrieval-augmented generation versus using it as a safety constraint. The authors make a bold, well-supported assertion that “parametric factual fallback is a liability” in high-stakes applications.
- The paper creates usable artifacts; PLR, with degradation curves (SHUFFLE, IRRELEVANT-k, REMOVE-GOLD), a three-tier validation process, and a reward framework for factual mode, will create something the community can validate and implement.
- Section 9 addresses five significant objections with thorough counterarguments. The authors do not dismiss parametric knowledge altogether, but they carefully constrain their arguments to high-stakes factual mode applications.

Weaknesses
- Although the three-tier architecture is described at a high level, it still leaves many relevant design questions unaddressed; for example, how does the frozen reasoning core interact with the dynamic knowledge layer when creating a response (cross attention, contextual retrievals, or some other means)? How does the system ensure proper enforcement of the boundary between factual and non-factual?
- The authors recognize that limited closed corpora will limit the coverage of factual claims and that under-evidence abstention is warranted as a consequence of the limited size of the evidence base. The authors do not address that in many real-world, high-stakes applications, the evidence base will be incomplete, highly contested, and/or changing rapidly.
- When retrieval is disabled, it will change the model input data distribution rather than limit the availability of evidence; thus, the performance of a model with retrieval disabled may be significantly lower than expected due to the changed input context (data distribution), rather than the removal of the ability for retrieval to create a fallback on the parametric knowledge.
- The authors do not address many of the concerns associated with separating Factual vs. Creative Modes as outlined in Section 7.2; who makes the determination of which type of response would occur (user, system, or application developer)? What happens when there is a question that has both factual and non-factual elements (e.g., a medical question posed in everyday conversation)? Incorrectly classifying mode could create either too much constraint (abstaining when unnecessary) or too little (parametric fallback in high-stakes situations), creating a very significant design issue.
- Some of the sections (i.e., Section 3 limitations of fine-tuning) are generic limitations with no unique insights. The paper would benefit from more ruthless editing to allow the unique and strongest arguments to stand out.

**Support:**

2

---

> ### Author Rebuttal · Authors · 2026-03-30
>
> We thank the reviewer for the thoughtful and balanced feedback, and for recognising the importance of reframing retrieval as a safety constraint rather than a performance enhancement.
>
> We agree that the interaction between components can be made clearer. The proposed Separation-of-Powers should be understood not as a rigid architectural pipeline, but as a runtime contract governing how outputs are authorised. Concretely, the reasoning core generates candidate responses, but in Factual Mode, these responses are only surfaced if they can be grounded in retrievable evidence and validated through an attribution layer; otherwise, the system abstains or expresses uncertainty. We will revise the manuscript to clarify this control flow more explicitly.
>
> This design is consistent with retrieval-augmented paradigms that decouple reasoning from knowledge storage, treating the model as a synthesiser rather than a knowledge base [Lewis20; Borgeaud22].
>
> Importantly, this abstraction does not assume a simple retrieve-then-generate paradigm. Many claims—particularly in clinical or scientific domains—require multi-hop reasoning across multiple pieces of evidence. Our framework accommodates this by allowing claims to be supported through joint entailment across retrieved documents, consistent with prior work on evidence aggregation and verification [Thorne18; Jiang23].
>
> Regarding contested or incomplete evidence, we fully agree that real-world evidence is often uncertain or conflicting. Our proposal does not assume that the evidence base is authoritative. Instead, it enforces epistemic transparency, ensuring that claims are grounded in explicit, inspectable sources. This aligns with broader calls for interpretable and accountable AI systems [Doshi-Velez17], and with real-world requirements for monitoring and auditability in clinical AI deployment [Sendak20].
>
> On PLR evaluation, we agree that naive retrieval toggling may introduce confounds. This motivates the use of structured retrieval degradation strategies to isolate parametric reliance without conflating it with general performance shifts. We will clarify this more explicitly.
>
> On mode switching, we agree that Factual Mode should not be interpreted as a brittle binary. In practice, it operates as a selective constraint, where factual spans are evidence-gated while non-factual spans remain unconstrained. We will refine this description accordingly.
>
> We appreciate the reviewer’s constructive feedback and will incorporate these clarifications to strengthen the manuscript.

---

> > ### Author Rebuttal · Reviewer_vVGp · 2026-04-04
> >
> > Thank you for the thoughtful rebuttal. The explanations solve my concerns regarding the separation-of-powers framework, retrieval grounding, and factual mode behavior. I encourage you to incorporate these in the revision for better persuasiveness. Nevertheless, some questions remain regarding conceptual precision, e.g., how does this framework handle contested evidence, and how can PLR evaluation avoid confounding? Therefore, I maintain my score.

---

### Decision · Program_Chairs · 2026-04-30

**Decision:**

Accept (regular)

**Comment:**

This paper offers an interesting take on the emerging problem of synthetic data and hallucination. By arguing for a strict divide between parametric reasoning (LLMs) and factual authorization (traceability back to agreed facts), it challenges not only the current "black-box" status quo of LLMs but strongly suggests that fact-grounded retrieval is the only way to solve this problem. This meta-review intentionally hand-waves over troubling implementation ambiguities and issues raised by the reviewers, all basically aligned in the spirit of "how can you actually do this practically?".